# A Methodology to Predict the Fatigue Life under Multi-Axial Loading of Carbon Fiber-Reinforced Polymer Composites Considering Anisotropic Mechanical Behavior

**DOI:** 10.3390/ma16051952

**Published:** 2023-02-27

**Authors:** Joeun Choi, Hyungtak Lee, Hyungyil Lee, Naksoo Kim

**Affiliations:** 1Department of Mechanical Engineering, Sogang University, Seoul 04107, Republic of Korea; 2Polymer R&D Team, GS Caltex R&D Center, Daejeon 34122, Republic of Korea

**Keywords:** fiber-reinforced polymer, injection molding process, anisotropy, multi-axial stress state, fatigue life prediction

## Abstract

Carbon fiber-reinforced polymers (CFRP) have been actively employed as lightweight materials; yet, evaluating the material’s reliability under multi-axis stress states is still challenging owing to their anisotropic nature. This paper investigates the fatigue failures of short carbon-fiber reinforced polyamide-6 (PA6-CF) and polypropylene (PP-CF) by analyzing the anisotropic behavior induced by the fiber orientation. The static and fatigue experiment and numerical analysis results of a one-way coupled injection molding structure have been obtained to develop the fatigue life prediction methodology. The maximum deviation between the experimental and calculated tensile results is 3.16%, indicating the accuracy of the numerical analysis model. The obtained data were utilized to develop the semi-empirical model based on the energy function, consisting of stress, strain, and triaxiality terms. Fiber breakage and matrix cracking occurred simultaneously during the fatigue fracture of PA6-CF. The PP-CF fiber was pulled out after matrix cracking due to weak interfacial bonding between the matrix and fiber. The reliability of the proposed model has been confirmed with high correlation coefficients of 98.1% and 97.9% for PA6-CF and PP-CF, respectively. In addition, the prediction percentage errors of the verification set for each material were 38.6% and 14.5%, respectively. Although the results of the verification specimen collected directly from the cross-member were included, the percentage error of PA6-CF was still relatively low at 38.6%. In conclusion, the developed model can predict the fatigue life of CFRPs, considering anisotropy and multi-axial stress states.

## 1. Introduction

Carbon fiber-reinforced polymer (CFRP) is widely utilized in various industries, such as automobiles and aerospace, for its high specific strength, corrosion resistance, and superior dimensional precision [1,2,3,4]. In addition, short fiber-reinforced composites manufactured through injection molding have noble productivity [5,6,7,8,9]. The material properties of CFRP can vary depending on the type of plastic. Thermosetting plastics have excellent mechanical properties and chemical resistance but have recycling difficulties and extended production time [10]. On the other hand, thermoplastic is flexible and reusable, but upon reheating, the physical properties can be reduced [11,12]. This study adopted polyamide-6 (PA6) and polypropylene (PP) thermoplastic resins as the CFRP matrix.

Despite CFRP’s remarkable advantages, the fiber-reinforced polymer (FRP) has been over-designed in the last decades to compensate for its high possibility of debonding and breakage at the fiber-matrix interface [13]. Furthermore, the complex short fiber orientation formed by the polymer flow in the injection molding process generates severe anisotropy of CFRP, making it challenging to secure design reliability compared to conventionally used metals [14,15,16,17].

Therefore, fatigue life prediction considering the anisotropic behavior induced by the fiber orientation is essential for enhancing the durability of CFRP parts and their economic potential. The alignment of the fibers caused by the polymer flow has a fatal effect on fatigue strength [18,19]. In particular, short carbon fiber compounds have a lower fatigue strength than long fiber-reinforced composites due to their shorter nominal fiber length [20,21]. Moreover, if the fiber orientation and principal stress in a specific region susceptible to breakage are transverse to each other, the fiber can be easily separated from the matrix, resulting in reduced static and cyclic mechanical properties [22]. Nevertheless, the utilization of short fibers is advantageous in the injection molding process, for their ease of application. In addition, short fiber is more likely to be selected for producing recycled fiber. Therefore, there is a need to analyze the fatigue behavior of short fiber-reinforced plastics.

Fiber orientation plays a crucial role in determining the anisotropy of injection-molded CFRP [23]. The surface layer near the cavity wall has a fiber orientation aligned in the direction of the polymer flow. In contrast, the fibers in the core region are aligned in the lateral direction of the flow [24,25,26,27]. However, for short FRP, the fibers are randomly distributed in the core region [28]. Still, the mechanical properties of the short FRP can be characterized from a macro-mechanical perspective using specific sample angles [29].

Many studies have utilized microcomputer tomography, experiments, and numerical simulations to analyze the anisotropic behavior of FRP [30,31,32,33,34,35]. For instance, Vincent et al. [30] developed an injection molding analysis model by computing fiber orientation tensors through two-dimensional optical image data. Meanwhile, studies on the durability characteristics of FRPs in complex stress states were also conducted [36,37,38,39]. In particular, Gude et al. proposed failure criteria through S-N curves obtained through multi-axis fatigue experiments of CFRP [36]. Furthermore, Tanaka et al. [39] conducted a study that simultaneously considered the notch effect and anisotropy by using single-edge notched plate specimens in different directions.

The fatigue fracture mechanisms of short fiber reinforced PA6 and PP have been investigated based on experimental and Scanning Electron Microscope (SEM) images [40,41,42,43]. PA6 fiber reinforcement showed local ductile fracture behavior at room temperature with little fiber pullout. In addition, cohesive deformation occured around the separated fibers due to the high interfacial bonding [40,41]. Meanwhile, the PP fiber reinforcement showed crack propagation due to fiber failure or pullout. Furthermore, the ductile fracture behavior is reduced as the fiber volume fraction is lowered [42,43].

As concerns the fatigue life prediction model, traditional analytical models, such as Manson–Coffin [44] and Smith Watson and Topper (SWT) [45], have been adopted to describe the anisotropy and complex stress effects [46,47,48,49,50,51,52,53,54]. Branco et al. [48] suggested the modified Manson–Coffin model capable of predicting fatigue life in complex stress states by adopting the equivalent strain energy density concept. Meanwhile, many semi-empirical models considering anisotropy have been developed by using angles or orientations in the model. In particular, Choi et al. [47] suggested the strain-based model, which accounts for the anisotropic behavior in multi-axial stress states by introducing an angle term of the specimens. In addition, Sonsino and Moosbrugger [55] predicted the fatigue behavior of glass FRP by calculating the mean stress and fiber orientation.

In summary, the durability of FRP can be described by calculating the orientation of individual fibers or by categorizing them into specific angles based on specimen cutting orientation. However, the above studies have limitations in that the fatigue life can be estimated only after the fiber orientation of the region of interest is known. In addition, few prior studies have evaluated anisotropy and various geometric effects as direct stress–strain relationships. Thus, there is a strong need to develop a methodology that can predict the cyclic fracture of CFRP based on numerical analysis and semi-experience models.

For the reasons mentioned above, a one-way coupled injection molding-structure numerical simulation and fatigue life model considering the anisotropy of CFRP in complex stress states have been developed. A Ramberg–Osgood flow stress model was selected to determine the anisotropic behavior. Model parameters were determined utilizing the tensile test results collected from different angles. In addition, the reduced strain closure (RSC) model was used to calculate the fiber orientation induced by the polymer flow, and the coefficient was derived via 3D X-ray CT (XCT) data. The fiber orientation mapping results derived from the finite volume method (FVM) were adopted as the initial material property of the finite element method (FEM). Finally, the FEM calculates the stress–strain and triaxiality. It was confirmed that all proposed procedures could be applied to PA6-CF and PP-CF and are not limited to specific materials.

Notch and complex shape specimens were collected from three angles (0° = injection direction, 45°, and 90°), and a fatigue test was conducted under various strain conditions. A semi-empirical fatigue life prediction model was developed based on the experimental and simulation results. The proposed model does not utilize fiber orientation angles or tensors in specific regions. Instead, the model developed through a combination of stress and strain terms predicts fatigue life based on differences in mechanical properties caused by the principal stress and fiber direction relationship. Accordingly, durability may be evaluated using only stress and strain generated by various load conditions of essential components in an operating environment. The proposed methodology allows designers to consider the anisotropic effect of CFRP components, assess vulnerability in advance, and pursue the convenience of varying the design without needing a prototype.

## 2. Materials and Methods

The fatigue life prediction of CFRP regarding anisotropy and multi-axial stress was performed through the procedure shown in Figure 1. First, static and cyclic experiments were conducted to characterize mechanical properties. In addition, the fiber orientation at 27 injection-molded plate regions was measured through XCT. The obtained image data were post-processed to calculate the short fiber orientation according to the polymer flow.

Secondly, the flow stress model coefficients for calculating the elastoplastic behavior were determined by reverse engineering the true stress–strain curve obtained in the tensile experiments. Furthermore, the RSC fiber orientation distribution parameter was derived from XCT data. Next, one-way coupled numerical analysis was modeled by combining the injection molding process and structural analysis utilizing the FVM and FEM. The injection molding parameters have been adopted based on the actual manufacturing process.

Finally, a semi-experiential fatigue life prediction model in the form of an energy function was developed based on the stress–strain-triaxiality relationship by evaluating the anisotropy and the multi-axial stress state effects. The value of each term used in the model was calculated from the one-way coupled numerical analysis results. The stress and strain used in the model were calculated in the selected stabilized hysteresis loop. As a result, the fatigue life of CFRP can be successfully predicted through the developed procedure.

### 2.1. Materials and Experimental Methods

Short fiber injection molded PA6-CF and PP-CF materials were selected to evaluate the mechanical properties of CFRP under a multi-axial stress state. Tensile and displacement control fatigue tests are conducted to analyze static and cyclic behavior. The uniaxial and notched specimens were machined based on the ASTM D 638 02a TYPE IV standard [50]. The fracture behavior due to stress and strain concentration occurring in the notch can be analyzed using two types of notched geometry. The dimensions of complex specimens were specified based on the numerical analysis results such that the specimen covers a wide range of triaxialities. Detailed information on the dimensions of the specimens used is reported in Figure 2. As illustrated, the notch and complex specimens were designated as “Type I” and “Type II” depending on their geometry. The experiment and numerical analysis results were described using the same denotator.

The verification specimen was obtained by partially cutting the injection-molded automobile cross-member. All specimens except the verification were machined on the injection molded plate at 0° (injection direction), 45°, and 90°. Furthermore, specimens were gathered 80 mm from the injection gate to exclude the effect of random fiber orientation due to unstable polymer flow near the injection gate. Figure 3 shows the direction and location of the machined specimen and the details of the verification specimen.

Tensile and fatigue tests were performed at room temperature (20 °C) and with a relative humidity of 50% utilizing the Instron 3367 and 8871 test systems, respectively. In addition, all tests were conducted after drying at 120 °C for 2 h in an oven dryer and cooling at room temperature for 30 min to remove moisture from the sample. In the case of ASTM D 638 and notched specimens, displacement was measured by an extensometer with a 25 mm gauge length. However, the complex specimens could not attach to the extensometer, so machine displacements have been employed. Tensile tests were carried out at a 10 mm/min tensile speed, and the parameters of the Ramberg–Osgood flow stress model described in Section 2.2 were derived using the obtained stress–strain curve. In addition, stress relaxation experiments were conducted to analyze the mechanical properties depending on the relaxing behavior. Fatigue tests were controlled by mean displacement and amplitude at 1 Hz frequency.

### 2.2. Numerical Analysis Methods

The fibers and matrix of the injection molded CFRPs are continuously damaged from exposure to repeated multi-axial stress. This phenomenon is intensified by complex stress states in the actual operating environment. Therefore, it is essential to accurately calculate the fiber orientation induced by the injection process and the stress–strain field in the fracture risk area. For this reason, a one-way coupled simulation of the injection process and the FEM was developed. The numerical analysis procedure is summarized in Figure 4.

First, an input file of Abaqus 2017, without material property data, is generated containing geometry, solid mesh, and boundary conditions. Simultaneously, Python scripts are used to calculate Lamberg–Osgood flow stress parameters from stress–strain data of tensile experiments. The RSC coefficient is also determined using XCT data. After that, the injection process analysis through FVM is performed by Autodesk Moldflow Insight (AMI). The material properties due to fiber directions are then mapped to the input file via Advanced Material Exchange Helius (AME). Finally, the stress–strain field is calculated through structural analysis by FEM, and the fatigue life of the CFRP part is predicted. The entire procedure utilizing the three commercial software is verified by comparing the calculated fatigue life and analysis results with the actual experimental results.

This research applied the RSC model to account for the increased tensile strength of the CFRP when the directions of carbon fiber and principal stress are the same. The RSC model considers the difference in orientation kinetics according to the location of the injection gate and accurately calculates the short-fiber orientation in complex geometries [56]. This model applies empirical factors and calculates fiber orientation by reducing the growth rate of the directional tensor without changing the rotation vector as follows:(1)$$\begin{array}{ll} \frac{Da_{ij}}{Dt} & =-\frac{1}{2}\left(\omega_{ik}a_{kj}-a_{ik}\omega_{kj}\right)\\ & +\frac{1}{2}\lambda \left({\dot{\gamma }}_{ik}a_{kj}+a_{ik}{\dot{\gamma }}_{kj}-2\left[a_{ijkl\dot{\gamma }kl}+\left(1-\kappa \right)\left(L_{ijkl}-M_{ijmn}a_{mnkl}\right)\right]{\dot{\gamma }}_{kl}\right)\\ & +2\kappa C_{I}\dot{\gamma }\left(\delta_{ij}-3a_{ij}\right) \end{array}$$Here, $a_{ij}$ is the fiber orientation tensor, $0.5w_{ij}$ the vorticity tensor, $0.5{\dot{\gamma }}_{ij}$ the deformation rate tensor, λ the particle shape coefficient, κ the scalar factor, L and M the fourth-order tensor, and $C_{I}$ the fiber interaction coefficient. The RSC factor between 0 and 1 is an empirical parameter, and the fiber alignment becomes similar to the polymer flow direction as the factor approaches 1.

The RSC parameters were optimized through reverse engineering by comparing fiber orientation calculated from XCT data with fiber orientation results from numerical simulations. Three steps are required to utilize the fiber orientation of the XCT data: (i) X-ray computed tomography, (ii) image data preprocessing through visualization, and (iii) fiber orientation calculation. First, XCT was conducted in three locations of the injection molded plate with dimensions of 2.5 × 2.5 × 1.96 mm^3^. Three different locations were selected to consider the diversity of fiber orientation. Secondly, for the preprocessing of the image data, the collected raw images have been reconstructed in 3D with lighting and transparency control and sectionalization with Dragonfly 2021. With the open_iA open-source tool for XCT datasets [57], the 3D data are converted to a combination of pixels. Finally, the fiber orientation tensor from the XCT data is calculated and compared with the Moldflow analysis results. The RSC parameters are updated until the difference in fiber orientation becomes less than 10% by the suggested procedure. The developed RSC parameter-determining procedure and the locations of the XCT photographing samples selected from the PA6-CF injection molded plates are described in Figure 5.

Injection process analyses were conducted using Moldflow commercial software with specified process variables and injection locations. Two models were developed using injection plates and automobile cross-member 3D-CAD step files. The simulations consisted of filling and packing processes, and the tetrahedra mesh was generated with 809,708 and 2,658,379 elements, respectively. A total of nine injection gate locations were defined concerning the location of the actual injection gates: one for the plate and eight for the cross-member. In addition, each process parameter used for the plate and the cross-member was designated in the process setting step. The variables related to temperature, filling, and packing used in the analysis are summarized in Table 1.

Material properties were interpolated into the Abaqus 2017 input file considering the fiber orientation calculated in the injection process analysis using the Helius commercial program. The Ramberg–Osgood flow stress model and the modified Hill’48 yield function were used to evaluate the specimens’ fiber orientation.
(2)$$\sigma_{eff}=\sqrt{\frac{{\left(\alpha \sigma_{11}-\beta \sigma_{22}\right)}^{2}+{\left(\beta \sigma_{22}-\beta \sigma_{33}\right)}^{2}+{\left(\beta \sigma_{33}-\alpha \sigma_{11}\right)}^{2}+6\left[{\left(\sigma_{12}\right)}^{2}+{\left(\sigma_{23}\right)}^{2}+{\left(\sigma_{31}\right)}^{2}\right]}{2}}$$
(3)$$\sigma =E^{1/n}K^{\left(n-1\right)/n}\varepsilon_{p,eff}{}^{1/n}$$

Here, E is an elastic modulus, K the strength coefficient, and n the hardening exponent. In addition, α and β are dependent parameters to $\lambda_{I}$ which define the first eigenvalue of the fiber orientation tensor. $\alpha_{m}$, $\beta_{m}$ and $\lambda_{m,I}$ are the reference values calculated in the Helius of the material.
(4)$$\sigma \left(\lambda_{I}\right)=\theta +\left[\frac{\left(\alpha_{m}-\theta \right)}{\left(\lambda_{m,I}-1/2\right)}\right]\left(\lambda_{I}-1/2\right),\;\beta \left(\lambda_{I}\right)=\theta +\left[\frac{\left(\beta_{m}-\theta \right)}{\left(\lambda_{m,I}-1/2\right)}\right]\left(\lambda_{I}-1/2\right)$$The material parameters have been determined by minimizing the differences between experimental and numerical load-displacement curves, as shown in Table 2.

Each specimen was modeled with 10,000 to 120,000 elements using the types C3D8R and C3D10, which are hexahedral and tetrahedral meshes. Fixed and displacement boundary conditions were assigned to apply experiment conditions. In addition, the rigid body constraint was given to the specimen part attached to the jig of the experimental machine. The boundary conditions adopted in injection and structure analysis are shown in Figure 6.

### 2.3. Fatigue Life Prediction Model

Predicting the fatigue life of short CFRPs that undergo complex stress conditions is challenging. In the case of short FRPs, fibers are randomly distributed at the center in the thickness direction of the product. Conversely, fibers are arranged with directionality on the product’s surface, resulting in considerable anisotropy. In addition, fibers exhibit excellent strength in the longitudinal direction but low mechanical properties in the transverse direction.

Many previous studies have been conducted using the concept of local fiber orientation, geometry, or deformation concentration in a fatigue life model to predict the fatigue life of short CFRPs. In this study, a semi-empirical model based on the strain–stress-based energy function has been proposed. The interaction between the fiber orientation and the principal stress determines the durability of the fragile part. Therefore, the fatigue life prediction model has been developed based on strain–stress relationships induced by the macroscopic fiber orientation, even if fiber orientation itself is not included in the model.

The CFRP’s semi-empirical fatigue life prediction model is developed based on the Manson–Coffin model, which relates the strain amplitude and the failure cycle [58].
(5)$$f\left(\varepsilon \right)=\frac{\left(\varepsilon_{p,\max}-\varepsilon_{p,\min}\right)}{2}=\Delta \varepsilon_{p}=A{\left(N_{f}\right)}^{c}$$

This model investigated the relationship between the plastic strain amplitude $\Delta \varepsilon_{p}$ and the fatigue fracture cycle $N_{f}$. The plastic strain amplitude is calculated by the max and min plastic strain, $\varepsilon_{p,\max}$ and $\Delta \varepsilon_{p,\min}$. Furthermore, two material constants, A and c, have been utilized to predict the failure cycle.

In addition, the Manson–Coffin–Basquin model has introduced elastic deformation and failure stress to the Manson–Coffin Model [59].
(6)$$\varepsilon_{a,t}=\varepsilon_{a,e}+\varepsilon_{a,p}=\frac{{\sigma^{\prime }}_{f}}{E}{\left(2N_{f}\right)}^{b}+{\varepsilon^{\prime }}_{f}{\left(2N_{f}\right)}^{c}$$

This model investigated the relationship between the total strain amplitude $\varepsilon_{a,t}$ and the fatigue fracture cycle $N_{f}$ by dividing it into elasticity and plastic strain amplitude, $\varepsilon_{a,e}$ and $\varepsilon_{a,p}$. The elastic strain amplitude can be figured with the fracture stress ${\sigma^{\prime }}_{f}$ divided elastic modulus E term with b, and the plastic strain amplitude can be calculated with the fracture strain ${\varepsilon^{\prime }}_{f}$ with c. *b* and *c* are the material constants.

The proposed model in this research has been developed by combining the above models and adding stress amplitudes and triaxiality terms.
(7)$$f\left(\varepsilon ,\sigma ,\eta \right)=\left(\frac{\varepsilon_{\max}-\varepsilon_{\min}}{2\varepsilon_{f,i}}\right){\left(\frac{\sigma_{\max,i}-\sigma_{\min,i}}{2\sigma_{f,i}}\right)}^{\ln\left(\frac{\sigma_{f,i}}{\sigma_{\max,i}}\right)}{\left(\eta_{i}\right)}^{\ln\left(1+\varepsilon_{f,i}\right)}=A{\left(N_{f}\right)}^{c}$$The significant difference of the proposed model is that it consists of the energy function as the product of stress and strain amplitude. The fracture values at the static experiments have been utilized for the failure stress $\sigma_{f,i}$ and strain $\varepsilon_{f,i}$ in the model. The first term allows the adoption of the model for CFRPs with various matrices. CFRP shows a difference in strain depending on the strength and rigidity of the base material. This study confirmed whether the life model could consider the difference in physical properties according to the matrix, using PA6-CF and PP-CF.

The second term is the stress amplitude term, which is calculated as half the difference between the maximum and minimum equivalent stresses, $\sigma_{\max,i}$ and $\sigma_{\min,i}$. In addition, by providing the maximum and failure stress to the exponential term in logarithmic form, the stress term is multiplied twice when the natural logarithm is taken on both sides. By multiplying the stress term twice, the varying stress differences induced by the directionality of the specimens can be effectively calculated. Thus, allowing consideration of the effect of anisotropy on fatigue life. Furthermore, the third term accounts for the relationship between the fracture strain and the triaxiality $\eta_{i}$, enabling consideration of the complex stress state. All stress, strain, and triaxiality were extracted from the numerical analysis model. Fatigue model constants A and c were determined by regression.

## 3. Results

### 3.1. Static and Fatigue Mechanical Properties

The tensile strength (TS), elastic modulus (E) and elongation at fracture ($\varepsilon_{f}$) have been summarized in Table 3. The true stress-strain curves obtained from uniaxial specimen tensile tests of PA6-CF and PP-CF materials are shown in Figure 7. In addition, the load-displacement curves of all specimens with FEM results have been reported in Section 3.2. Static and fatigue tests were conducted on both CFRPs by cutting specimens at 0° (injection direction), 45°, and 90°. The static experiment results can confirm the effect of anisotropy and triaxiality on the material properties of the CFRPs. PA6-CF has higher tensile strength than PP-CF but breaks at a lower strain. All CFRPs showed high tensile strength at 0°, followed by 45° and 90°. As the triaxiality increased, the fracture stress and strain decreased. All static test results were repeated three times, and then the median value was used.

The true stress–time curves obtained from the stress relaxation tests of PA6-CF and PP-CF are shown in Figure 8. Stress losses occurred more rapidly to PA6-CF than to PP-CF. This rapid reduction can be inferred from the fact that the interfacial bonding between the matrix and the fiber is higher in PA6-CF, resulting in quicker stress dispersion. The strength loss was relatively low when both CFRP were 0° than 90° in the results according to the three angles.

The true stress–strain values of the uniaxial specimen were calculated directly using the experiment results, and for other specimens, the stress–strain was obtained from FEM. The fatigue test conditions and results are presented in Table 4 and Table 5, respectively. All values were extracted from a stabilized hysteresis loop with a difference of less than 5% from the failure stress–strain hysteresis loop.

SEM images were taken at PP-CF and PA6-CF fracture cross sections, as shown in Figure 9. Although both materials had carbon fibers with a volume fraction of 20%, they showed a clear difference in fracture characteristics. In PP-CF, many fibers were pulled out due to the weak interfacial bonding between the matrix and the fiber. Conversely, PA6-CF had a relatively high interfacial bonding showing cohesive failure without many fiber pullouts. The fracture behavior of PP-CF occurred in the order of matrix cracking, fiber breakage, and fiber pullout, while fiber breakage and matrix cracking occurred simultaneously for PA6-CF.

### 3.2. Numerical Analysis Results

The XCT data were obtained to optimize the RSC parameter, and the fiber orientations have been visually confirmed. XCT data of the PA6-CF injection molded plate is shown in Figure 10. Black, gray, and white represent the CFRP’s void, matrix, and fiber, respectively. Figure 10a shows a full 3D image of single XCT data, and Figure 10b reveals the fiber distribution along the thickness and injection directions. The fibers in the surface layer are aligned with the injection direction. In contrast, the fibers are randomly distributed and appear as dots in the core layer. Figure 10c is an enlarged image of the surface layer fiber orientation viewed normal to the thickness direction.

The image is converted into coordinate data in pixel units through the Open_Ai open source tool. Three-dimensional data are stored as a three-dimensional coordinate system with a unique ID per pixel. The data of each pixel ID consist of Node ID and fiber vector. Additionally, fiber orientation vectors of T*_xx_*, T*_yy_*, T*_zz_*, T*_xy_*, T*_xz_*, and T*_yz_* are calculated for each Node. The fiber orientation vector is defined with the fiber length *l* with orientation angles as follows:(8)$${\tilde{v}}_{f}=\left(\begin{array}{c} v_{f,x}\\ v_{f,y}\\ v_{f,z} \end{array}\right)=l\left(\begin{array}{c} \sin\alpha \cos\beta \\ \sin\alpha \sin\beta \\ \cos\alpha \end{array}\right)$$

Figure 11a shows the 9 data extraction locations and compares the fiber orientation vector between the image and injection analysis results. Fiber orientation results show high accuracy with a 3.96%, 5.27%, and 4.38% difference in the principal direction. Fiber orientation tensor components were automatically calculated using fiber vectors at each point of the injection plate. The optimal RSC parameter was selected as 0.472, and the resulting fiber orientation tensor components are shown in Figure 11b. A total of 61 iterative calculations were made to generate the results.

The injection process simulations were performed using the obtained optimal RSC parameter and the proposed one-way coupled analysis method. The fiber orientation distribution results of the injection molding process analysis through FVM are shown in Figure 12. For the injection plate, the fiber distribution of the surface layers of both CFRPs is shown at the top. In addition, the results in the middle layer of the injection plate are shown at the bottom. The results of the injection cross-member are displayed by dividing them into top and bottom views on the right.

The fiber orientation of the middle layer is more consistent with the injection direction when compared to the surface layer. Since the center region is filled first and thus exposed to molten polymer flow the longest, the middle layer fiber orientation is closer to 1. In the surface layer, the overall fiber orientation value is 0.73. Meanwhile, PA6-CF showed a larger fiber orientation tensor than PP-CF. This phenomenon explains why the anisotropy of PA6-CF was more significant in static and fatigue tests. In addition, it was found that the rib segments of the cross-member were more susceptible to fracture due to their considerable difference in fiber orientation. Fiber orientation mapping was performed using AME to consider the anisotropy in structural analysis. Figure 13 shows the mapping results of the 90^o^ cases of each specimen and the cross-member.

The FEM model calculated stress, strain, and triaxiality based on the applied FVM fi-ber orientation results. The one-way coupled analysis was verified by comparing the load-displacement curve obtained from the static test with the results from the structural analysis. The triaxiality calculated through structural analysis is shown in Table 4 and Table 5 with fatigue experiment results. Furthermore, the comparison results for all specimens are reported in Figure 14, Figure 15, and Figure 16, respectively. The minimum and maximum average deviation were calculated to be 1.04% and 3.16% by comparing the area integration of experiment–analysis curves, supporting the reliability of the analysis model. In addition, it was confirmed that each specimen’s most prominent stress position and fracture position were identical. In conclusion, it was demonstrated that failure of CFRP parts due to stress concentration could be predicted through injection-structure one-way couped numerical analysis method.

The experiment and numerical analysis results showed very high agreement. The differences in area integral beneath the experiment–simulation curves by material, geometry, and angle are summarized in Table 6.

### 3.3. Fatigue Life Prediction

A semi-empirical fatigue life prediction model is developed based on the fatigue test results at varying angles of five different geometries. The model constants of the life model were derived from 29 datasets for both PA6-CF and PP-CF, and a verification set was constructed using the four test results of the complex specimen. In the case of PA6-CF, the fatigue life predictability of the actual injection product was reviewed through two types of cross-member parts. The model coefficients *A* and *c* were derived using the least squares method and are reported in Table 7. The regression line and stress–strain-triaxiality-based energy function-to-failure cycle graphs calculated by the obtained model constants are shown in Figure 17 and Figure 18, respectively.

## 4. Discussion

To develop a fatigue life prediction model for short CFRP, experimental and numerical analyses of PA6-CF and PP-CF were conducted. Anisotropy and exposure to complex stress state significantly affected CFRP’s static and cyclic mechanical properties. In particular, it was found that when a load was applied in a direction consistent with the fiber arrangement, the tensile strength significantly increased. In addition, higher triaxiality resulted in a more complicated fracture pattern and shorter fatigue life, making it difficult to predict the life of injection molded CFRPs. This study employed a triaxiality range of 0.333 to 0.655 to investigate the multi-axial stress state. Furthermore, it was confirmed that the triaxiality was higher when the load was applied in the opposite direction to the fiber arrangement with the same geometry specimen.

The effect of anisotropy caused by fiber orientation can be explained through the experimental results of the directionally machined specimens. In the case of static tests, the highest tensile strength was shown when the load was applied similar to the injection direction. PA6-CF showed a more severe anisotropy than PP-CF. This phenomenon occurs because PA6-CF’s fiber orientation is more consistent with the injection direction, as shown in the numerical analysis results.

One-way coupled injection-structure analysis was conducted using FVM and FEM. The FVM analysis model was developed using the RSC parameter optimized from XCT data to consider anisotropy accurately. A widely used Ramberg-Osgood model in CFRP was adopted to reflect the anisotropic effects of fiber orientation. In addition, the FVM results were mapped to the structural analytical model to provide the material property data for stress–strain calculation.

The FEM model was finally utilized to investigate stress, strain, and triaxiality and to evaluate stress concentrations. The load-displacement curves of the structural analysis for all specimens were compared with the experiments, and the maximum mean deviation was 3.16%, demonstrating the analytical model’s reliability. Furthermore, the actual fracture locations and the calculated stress concentration points of the cross-member specimens were consistent, thus, indicating the possibility of applying the proposed one-way analytical model for complex-shaped injection products.

A developed fatigue life prediction model introduces a strain–stress-triaxiality-based energy function considering both the anisotropy and complex stress states. Although the model does not contain a direct fiber orientation term, fatigue life can be predicted using the fiber orientation-induced stress difference. This is because anisotropy due to fiber orientation causes a difference in generated stress.

Therefore, the proposed model multiplies the stress term twice by taking the natural logarithm on both sides of the equation. This stress term allows the influence of the fiber to be implied in the model. In addition, damage due to strain and stress concentration in the complex stress state was predicted by adding a term consisting of a triaxiality and fracture strain. The developed semi-empirical fatigue life prediction results showed correlation coefficients (*R*^2^) of 0.9813 and 0.9797, respectively, showing PA6-CF and PP-CF, respectively, proving the model’s reliability.

The proposed fatigue life prediction procedure is highly accurate and can be widely used in designing CFRP injection molding products of complex shapes. The proposed methodology’s advantage lies in the fact that the micro-local fiber orientation does not have to be repeatedly analyzed when predicting the mechanical properties of injection molded products. In addition, this research presents the potential for designers to predict fatigue life without using prototypes through one-way coupled numerical analysis and lab-scale experiments. However, future research is still required to evaluate the proposed life prediction model further. By applying this procedure to design actual injection products and to a broader range of CFRP materials, a more universal and validated form of the proposed model could be derived.

## 5. Concluding Remarks

This study predicted the fatigue life considering the anisotropy and complex stress states of two CFRPs, PA6-CF and PP-CF. The concluding remarks are as follows:The tensile strength of a material increases as a load is applied in the same direction as the fiber orientation. In addition, the PA6-CF shows a more significant anisotropy than the PP-CF.A one-way coupled injection-structural analysis model was developed, and fiber orientation along the polymer flow was considered through XCT photography and the Ramberg–Osgood model.The semi-empirical fatigue life prediction model based on the energy function showed high prediction accuracy of PA6-CF and PP-CF with a correlation coefficient of 0.9813 and 0.9797, respectively.The developed fatigue life prediction methodology allows designers to reduce trial and error when designing products of complex geometries composed of injection molded short CFRPs and effectively predict fatigue life.

## Figures and Tables

**Figure 1 materials-16-01952-f001:**
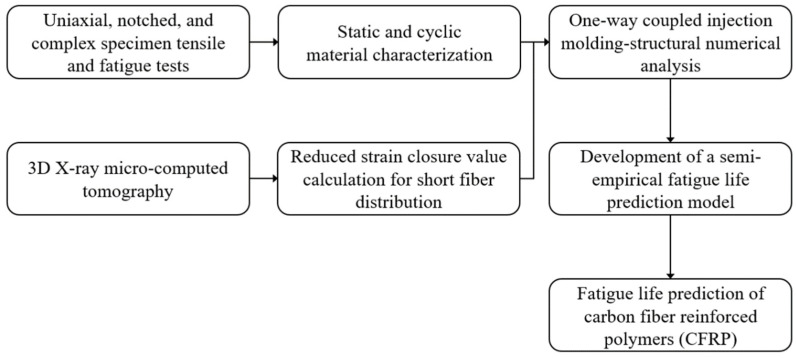
The procedure of fatigue life prediction for CFRP.

**Figure 2 materials-16-01952-f002:**
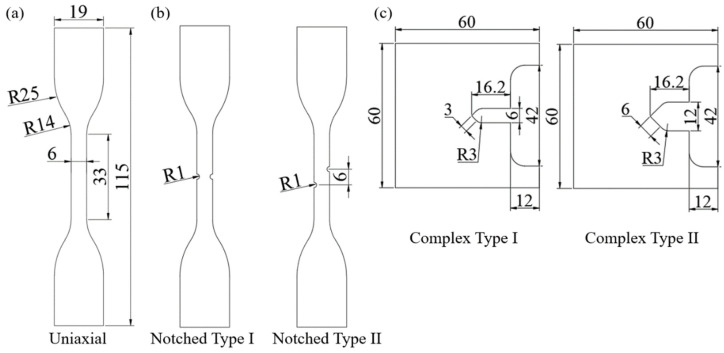
Dimensional specifications of test specimens denoted in mm. (**a**) ASTM-D638 Type IV standard uniaxial specimen. (**b**) Notched specimens. (**c**) Complex specimens.

**Figure 3 materials-16-01952-f003:**
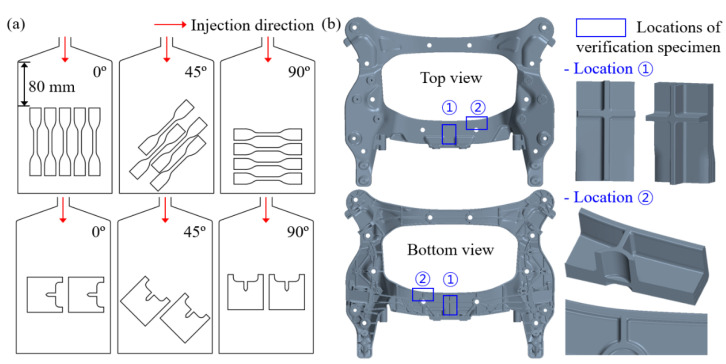
(**a**) Direction and location of specimens cut from injection molded plates. (**b**) Locations and geometries of verification specimens in injection-molded automobile cross-member components.

**Figure 4 materials-16-01952-f004:**
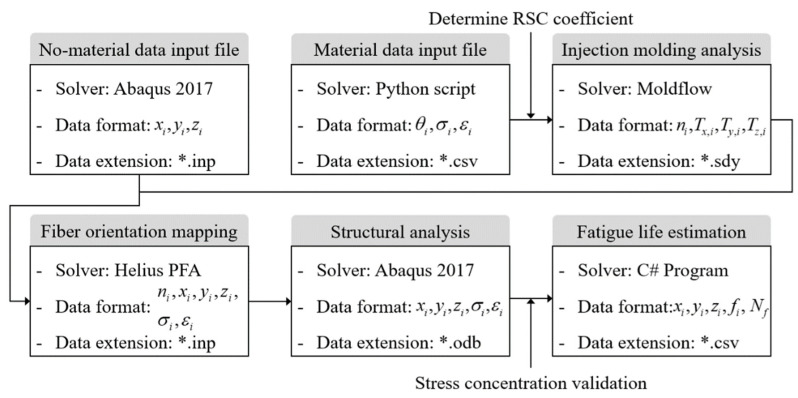
The one-way coupled injection molding-structure numerical analysis procedure for fatigue life prediction of CFRPs.

**Figure 5 materials-16-01952-f005:**
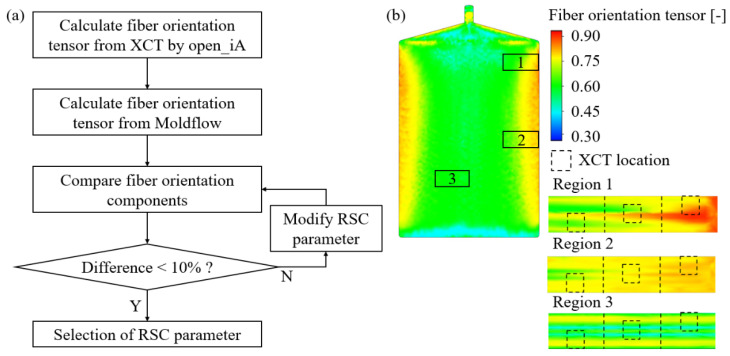
(**a**) The procedure to determine the RSC parameter by comparison of fiber orientation tensor obtained from XCT data and Moldflow simulation results. (**b**) The locations of XCT samples in the PA6-CF injection molded plate.

**Figure 6 materials-16-01952-f006:**
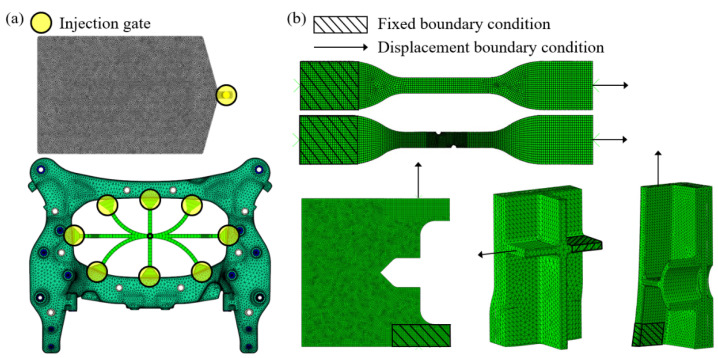
Boundary conditions and simulation models. (**a**) The injection gate and tetrahedra mesh for the injection molding simulation. (**b**) Mechanical boundary conditions for structural analysis.

**Figure 7 materials-16-01952-f007:**
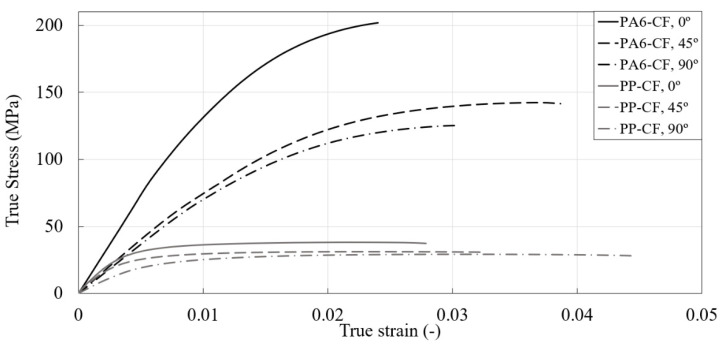
True stress and strain curves of the PA6-CF and PP-CF materials along 0° (injection direction), 45°, and 90°.

**Figure 8 materials-16-01952-f008:**
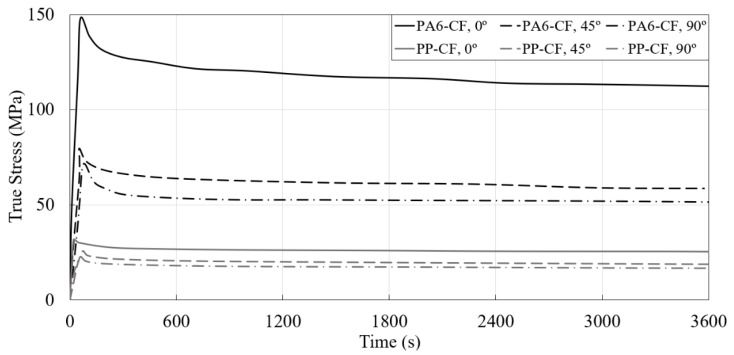
True stress–time curves of the PA6-CF and PP-CF obtained from the stress relaxation tests.

**Figure 9 materials-16-01952-f009:**
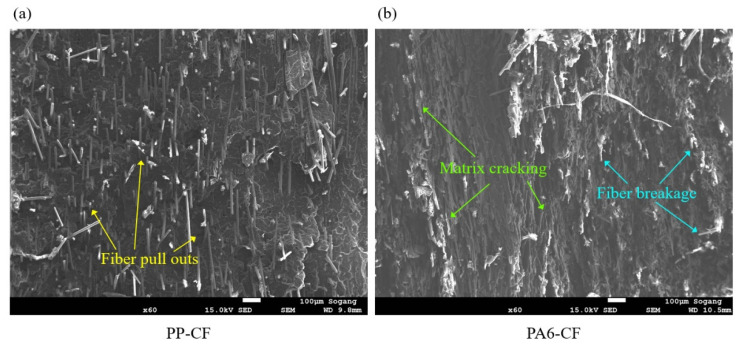
SEM images on fracture cross-sectional surface of (**a**) PP-CF and (**b**) PA6-CF.

**Figure 10 materials-16-01952-f010:**
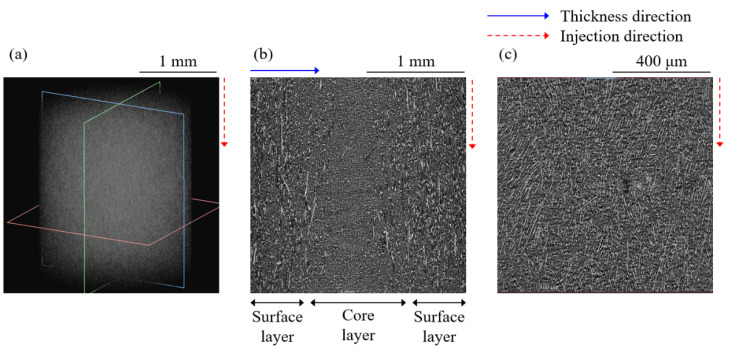
The XCT images from PA6-CF injection molded plate. Fibers and matrices appear as white and grey elements. (**a**) Full 3D view of a single XCT data. (**b**) Surface and core layer along thickness and injection direction. (**c**) Surface layer view from normal to the thickness direction.

**Figure 11 materials-16-01952-f011:**
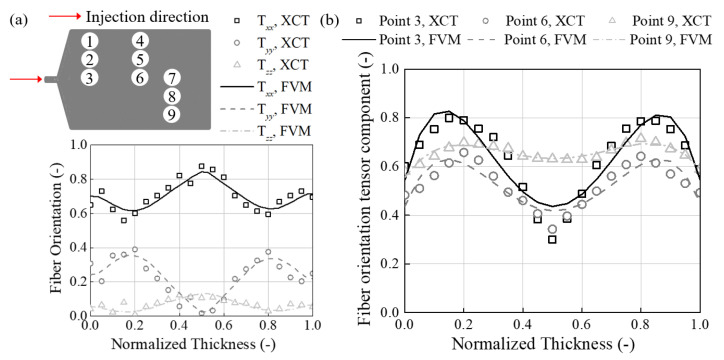
(**a**) The analysis points on the injection molded plate and the fiber orientation vectors on point 7. (**b**) The fiber orientation tensor component results on points 3, 6, and 9.

**Figure 12 materials-16-01952-f012:**
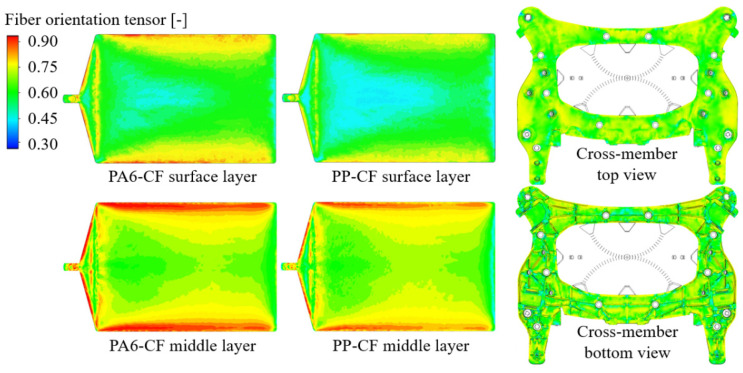
Fiber orientation distribution of surface and middle layer of injection molded plate, and cross-member.

**Figure 13 materials-16-01952-f013:**
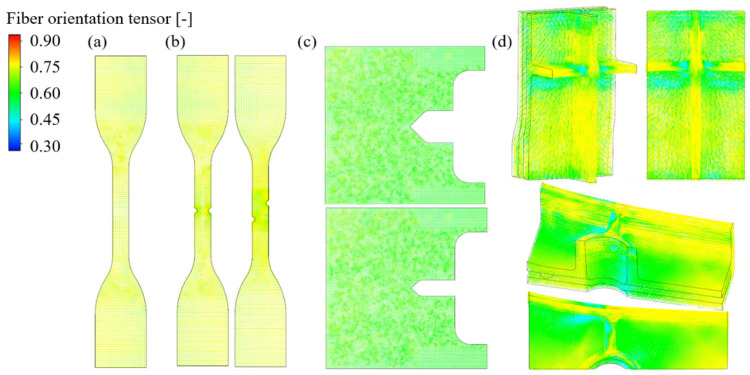
The mapping results of the (**a**) uniaxial, (**b**) notched, (**c**) complex, and (**d**) cross-member specimens.

**Figure 14 materials-16-01952-f014:**
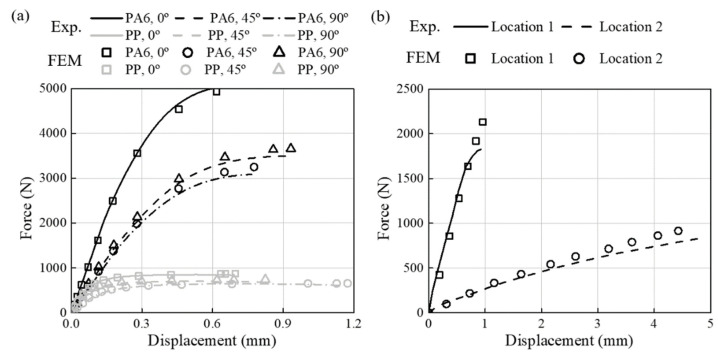
Load-displacement curves (line: experimental data, scatter: FEM results). (**a**) PA6-CF and PP-CF materials for the uniaxial specimens. (**b**) Cross-member specimens.

**Figure 15 materials-16-01952-f015:**
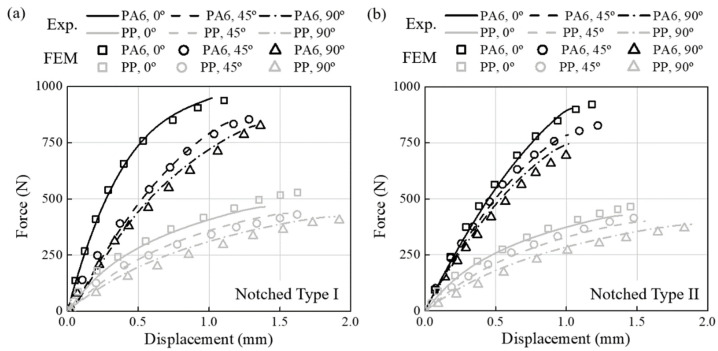
Load-displacement curves of PA6-CF and PP-CF materials (line: experimental data, scatter: FEM results). (**a**) The notched Type I. (**b**) The notched Type II.

**Figure 16 materials-16-01952-f016:**
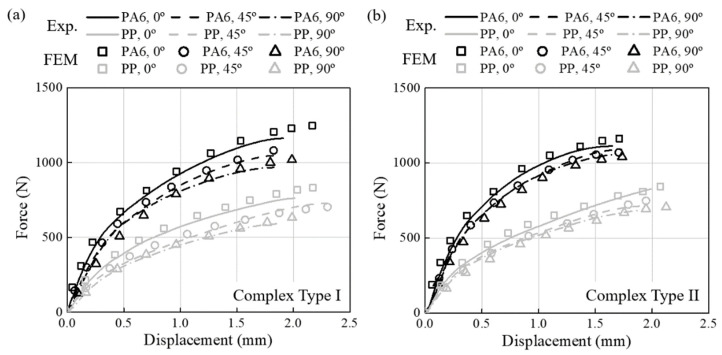
Load-displacement curves of PA6-CF and PP-CF materials (line: experimental data, scatter: FEM results). (**a**) The complex Type I. (**b**) The complex type II.

**Figure 17 materials-16-01952-f017:**
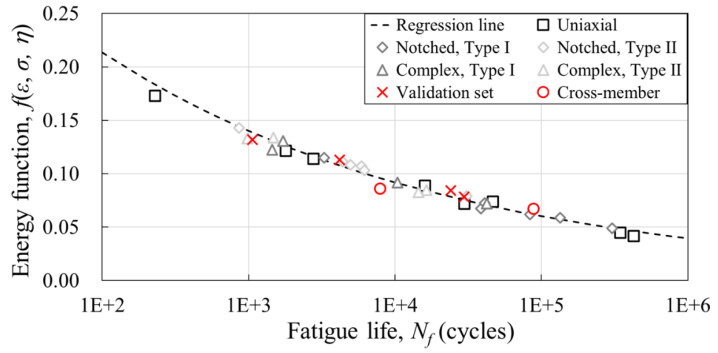
Energy function vs. fatigue life graph with regression line of the PA6-CF (log-linear scale).

**Figure 18 materials-16-01952-f018:**
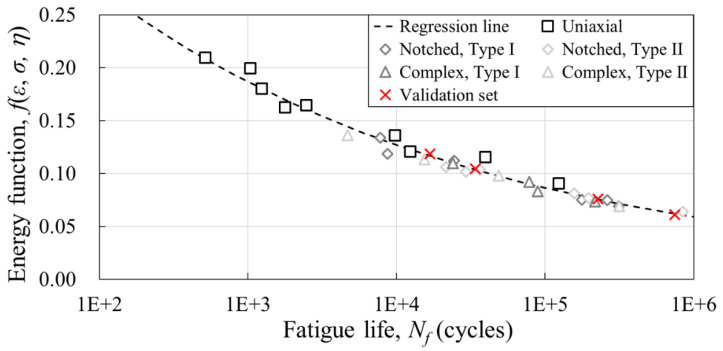
Energy function vs. fatigue life graph with regression line of the PP-CF (log-linear scale).

**Table 1 materials-16-01952-t001:** Process parameter setting values in the injection analysis for the plate and cross-member.

Process Parameters	Plate	Cross-Member
Mold surface temperature [°C]	85	
Melt temperature [°C]	285	
Switch-over by %volume filled [%]	99	
Injection time [s]	2	20
Cooling time [s]	20	140

**Table 2 materials-16-01952-t002:** Model constants for the Ramberg–Osgood flow stress model and a modified Hill’48 yield function.

Materials	K [MPa]	n [−]	$\alpha_{m}$ [−]	$\beta_{m}$ [−]	$\lambda_{m,I}$ [−]	$E_{m}$ [GPa]	$E_{f}$ [GPa]
PA6-CF	95.07	11.56	1.77	1.14	0.85	4.92	136.57
PP-CF	18.27	12.60	2.76	1.94	0.85	1.01	82.55

**Table 3 materials-16-01952-t003:** Mechanical properties of PA6-CF and PP-CF.

Material	Specimen Angle [°]	TS [MPa]	E [GPa]	$\varepsilon_{f}$ [−]
PA6-CF	0	201.9	14.7	0.024
	45	141.6	7.5	0.038
	90	125.1	6.8	0.030
PP-CF	0	37.2	8.3	0.027
	45	31.1	7.1	0.032
	90	27.8	4.3	0.044

**Table 4 materials-16-01952-t004:** Summary of fatigue experiments on the uniaxial, notched, complex, and cross-member specimens of PA6-CF.

Geometry	Triaxiality [η]	Specimen Angle [°]	$\varepsilon_{\max}$ [−]	$\varepsilon_{\min}$ [−]	${\left(\sigma_{vM}\right)}_{\max}$ [MPa]	${\left(\sigma_{vM}\right)}_{\min}$ [MPa]	*N_f_* [cycles]
Uniaxial	0.333	0	0.012	0.003	150.1	48.4	1800
			0.013	0.007	157.7	103.8	46,700
			0.017	0.007	183.1	103.8	230
		45	0.011	0.004	75.1	29.0	>10^6^
			0.013	0.007	92.2	58.4	428,500
			0.017	0.007	112.5	58.4	16,100
		90	0.012	0.003	80.9	25.1	29,800
			0.013	0.007	86.8	53.5	346,900
			0.017	0.007	103.2	53.5	1790
Notched Type I	0.408	0	0.116	0.087	300.2	275.8	83,400
			0.116	0.058	300.2	227.5	38,600
	0.475	45	0.163	0.108	210.4	172.9	134,000
			0.163	0.072	210.4	125.6	3280
	0.481	90	0.091	0.046	184.3	106.4	41,000
			0.091	0.057	184.3	128.3	304,000
Notched Type II	0.393	0	0.111	0.069	288.7	210.6	30,900
			0.111	0.055	288.7	176.6	4400
	0.469	45	0.143	0.067	223.5	164.2	4980
			0.143	0.072	223.5	138.1	5900
	0.498	90	0.180	0.113	228.3	163.6	6200
			0.180	0.090	228.3	136.5	860
Complex Type I	0.555	0	0.132	0.093	266.8	228.9	10,400
			0.132	0.078	226.8	214.1	1720
	0.612	45	0.159	0.114	238.6	207.6	29,600
			0.159	0.129	230.3	187.8	1450
	0.641	90	0.215	0.158	230.3	206.5	42,300
			0.215	0.129	230.3	187.8	4200
Complex Type II	0.544	0	0.107	0.071	268.5	233.2	16,400
			0.107	0.053	268.5	209.6	980
	0.602	45	0.139	0.092	237.5	207.9	24,000
			0.139	0.069	237.5	180.2	1480
	0.655	90	0.123	0.082	206.7	176.8	14,500
			0.123	0.061	206.7	151.2	1060
Location I	0.529	-	0.172	0.094	198.9	122.4	88,700
Location II	0.601	-	0.338	0.193	331.1	220.6	7960

**Table 5 materials-16-01952-t005:** Summary of fatigue experiments on the uniaxial, notched, and complex specimens of PP-CF.

Geometry	Triaxiality [η]	Specimen Angle [°]	$\varepsilon_{\max}$ [−]	$\varepsilon_{\min}$ [−]	${\left(\sigma_{vM}\right)}_{\max}$ [MPa]	${\left(\sigma_{vM}\right)}_{\min}$ [MPa]	*N_f_* [Cycles]
Uniaxial	0.333	0	0.014	0.004	37.3	29.9	1780
			0.016	0.004	37.6	29.9	1040
			0.017	0.006	37.8	32.9	520
		45	0.013	0.004	30.7	23.8	9780
			0.016	0.004	31.1	23.8	1240
			0.017	0.006	31.1	27.1	2480
		90	0.013	0.004	26.8	17.3	123,800
			0.016	0.004	27.6	17.3	12,400
			0.017	0.007	28.0	21.6	39,800
Notched Type I	0.379	0	0.152	0.102	161.7	131.1	262,400
			0.152	0.076	161.7	114.6	8700
	0.389	45	0.163	0.109	147.4	122.3	312,600
			0.163	0.081	147.4	104.2	24,500
	0.404	90	0.126	0.049	115.3	56.0	7800
			0.126	0.078	115.3	82.2	176,400
Notched Type II	0.376	0	0.121	0.073	145.3	112.6	196,000
			0.121	0.061	145.3	100.9	29,400
	0.421	45	0.146	0.087	141.8	101.6	156,700
			0.146	0.073	141.8	88.5	21,600
	0.435	90	0.145	0.100	123.6	97.1	846,200
			0.145	0.078	123.6	80.0	36,500
Complex Type I	0.467	0	0.137	0.082	154.3	126.3	89,500
			0.137	0.064	154.3	110.4	16,750
	0.481	45	0.165	0.110	140.0	120.0	746,500
			0.165	0.088	140.0	107.0	78,400
	0.512	90	0.213	0.142	134.9	114.2	216,500
			0.213	0.113	134.9	102.0	23,900
Complex Type II	0.467	0	0.134	0.089	153.0	132.4	314,500
			0.134	0.071	153.0	119.4	34,000
	0.479	45	0.177	0.118	140.8	118.2	226,500
			0.177	0.095	140.8	105.8	15,400
	0.501	90	0.156	0.094	176.2	131.5	48,600
			0.156	0.073	176.2	110.2	4700

**Table 6 materials-16-01952-t006:** The deviation between the experimental and simulation load-displacement curves.

Materials	Geometry	Specimen Angle [°]	Deviation [%]
PA6-CF	Uniaxial	0	1.04
		45	2.05
		90	2.68
	Notched Type I	0	2.08
		45	2.79
		90	1.38
	Notched Type II	0	2.81
		45	2.37
		90	2.51
	Complex Type I	0	2.64
		45	2.87
		90	2.92
	Complex Type II	0	2.58
		45	1.73
		90	2.51
	Cross-member location 1	-	2.94
	Cross-member location 2	-	3.16
PP-CF	Uniaxial	0	1.28
		45	2.78
		90	2.64
	Notched Type I	0	1.13
		45	2.41
		90	2.95
	Notched Type II	0	2.43
		45	2.55
		90	2.89
	Complex Type I	0	2.42
		45	1.68
		90	1.27
	Complex Type II	0	2.61
		45	1.42
		90	2.88

**Table 7 materials-16-01952-t007:** Model constants for the proposed semi-empirical fatigue life prediction models.

Materials	A [−]	c [−]
PA6-CF	0.4958	−0.183
PP-CF	0.5915	−0.167

## Data Availability

Not applicable.

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
