# Peer review of "A Methodology to Predict the Fatigue Life under Multi-Axial Loading of Carbon Fiber-Reinforced Polymer Composites Considering Anisotropic Mechanical Behavior"

_materials, 2023, doi:10.3390/ma16051952_

Round 1
Reviewer 1 Report
Choi et al. have presented the manuscript titled: A Methodology to Predict the Multi-axial Fatigue Life of Carbon Fiber-Reinforced Polymer Composites Considering Anisotropic Mechanical Behavior. Overall presentation of the article is good. I have few suggestions for the authors about this article.
1. Mechanical properties of molded PA6-CF and PP-CF materials depends upon the stretching ability and relaxing behaviors, for such purpose these properties are important to measure.
2. Moreover the mechanical properties of such materials also dependent on the porosity and density, is it possible for the authors to show the SEM images of such materials cross-sectional images?
Overall the experimental and theoretical discussion is good, article can be published following the above two points to enhance the experimental section as well.
Reviewer 2 Report
In this paper, the fatigue properties of short carbon fiber reinforced resin composites are systematically studied by experimental tests and FEA. The experimental data are fully analyzed and highly consistent with the finite element simulation results. The paper is well designed with rich data analysis, and some research results are important to understand the fatigue performance of materials. To further improve the quality of the paper, you can consider the following comments.
1. Abstract, please provide the quantitative results related to the fatigue life of CFRP. In addition, the analysis on the fatigue damage mechanism of CFRP should be supplemented.
2. Introduction, this paper adopted the thermoplastic resin (polyamide-6 (PA6) and polypropylene (PP)) as the resin matrix to investigate the fatigue performance. However, the authors didn’t summarize the obvious differences between thermosetting and thermoplastic resins composite when analyzing the advantage and application in the first paragraph. For thermosetting resin matrix composites, it has the mature preparation process, excellent mechanical properties, corrosion and creep resistances. However, the higher price and large brittleness of thermosetting resins may be the main problems. In contrast, thermoplastic resin matrix composites are green and recyclable with lower price, higher toughness. However, high viscosity and low fluidity of resin may lead to the poor interface bonding performance. It is suggested that the authors consider the advantage differences of two resin matrix composites and provide an systematic summary in paragraph one through reviewing the following research: Composite Structures, 2022, 293, 115719. Composite Structures. 2021, 261: 113285. Polymers, 2022, 14, 2953.
3. The authors claims “short carbon fiber compounds have a lower fatigue strength than long fiber-reinforced composites due to their shorter nominal fiber 40 length [17,18].”, So why do you use short carbon fiber in this paper? It is well known that fiber mainly depends on its high properties, so long fiber has higher tensile strength.
4. Please further add relevant literature summary on fatigue damage mechanism and fatigue evolution process of composite materials. In addition, what is the fatigue performance of pa6 and PP resin matrix composites? Is there any relevant research work?
5. Why do you conduct the test of notched specimens in Figure 2? The notched specimens may lead to initial stress concentration due to irregular shape, which will add some initial defects to the material and reduce its strength.
6. In the 3.1 part, the authors are suggested to give some tensile test data, such as tensile strength, tensile modulus, elongation at break and other basic information.
7. It can be found that the agreement between the experimental results and the finite element simulation results is very good. It is suggested that the authors provide a summary table to give the agreement degree data between the experimental and finite element results.
8. The conclusion part is suggested to be further condensed to contain the 3-4 key information related to this paper.
9. The data in the table in the appendix are important. It is suggested to move them to the main text.
Round 2
Reviewer 2 Report
Accepted.